# What Can Inflammation Tell Us about Therapeutic Strategies for Parkinson’s Disease?

**DOI:** 10.3390/ijms25031641

**Published:** 2024-01-29

**Authors:** Jinsong Xue, Keju Tao, Weijia Wang, Xiaofei Wang

**Affiliations:** School of Biology, Food and Environment, Hefei University, Hefei 230601, China; 13329103958@163.com (K.T.); 18110818256@163.com (W.W.)

**Keywords:** Parkinson’s disease, inflammation, mechanisms, therapies

## Abstract

Parkinson’s disease (PD) is a common neurodegenerative disorder with a complicated etiology and pathogenesis. α-Synuclein aggregation, dopaminergic (DA) neuron loss, mitochondrial injury, oxidative stress, and inflammation are involved in the process of PD. Neuroinflammation has been recognized as a key element in the initiation and progression of PD. In this review, we summarize the inflammatory response and pathogenic mechanisms of PD. Additionally, we describe the potential anti-inflammatory therapies, including nod-like receptor pyrin domain containing protein 3 (NLRP3) inflammasome inhibition, nuclear factor κB (NF-κB) inhibition, microglia inhibition, astrocyte inhibition, nicotinamide adenine dinucleotide phosphate (NADPH) oxidase inhibition, the peroxisome proliferator-activated receptor γ (PPARγ) agonist, targeting the mitogen-activated protein kinase (MAPK) pathway, targeting the adenosine monophosphate-activated protein kinase (AMPK)-dependent pathway, targeting α-synuclein, targeting miRNA, acupuncture, and exercise. The review focuses on inflammation and will help in designing new prevention strategies for PD.

## 1. Introduction

Motor symptoms like bradykinesia, muscle rigidity, resting tremor, and postural instability are the hallmark of Parkinson’s disease (PD), a chronic and progressive neurological disorder. The degeneration of dopaminergic (DA) neurons in the substantia nigra (SN) and aggregation of α-synuclein are the cardinal indicators of the disorder [1]. Studies have verified that PD pathogenesis is linked to a variety of mechanisms, such as neuroinflammation, α-synuclein aggregation, lysosomal autophagy system dysfunction, mitochondrial damage, a vesicle transport defect, and gut microbiome dysbiosis [2,3,4]. Neuroinflammation is a common pathway among several pathogeneses and plays a key factor in initiating and developing PD pathology [5,6].

Different stimuli could be responsible for the detected brain inflammation in PD. These include genetic factors, environmental toxins, heavy metals, head trauma, and microorganisms (e.g., viruses and bacteria) [2,3,7,8]. For example, several genetic studies have discovered a correlation between immune-related genes like bone marrow stromal cell antigen 1 (*BST1*) and the human leukocyte antigen (*HLA*) and PD [9,10,11,12]. Microglial activation resulting from 1-methyl-4-phenyl-1,2,3,6-tetrahydropyridine (MPTP) exposure has been found to cause irreversible neuronal damage and parkinsonian syndromes, even after 16 years [13]. Rotenone was reported to induce PD-like phenotypes via impairing the immune responses [14]. It has been demonstrated that abnormal iron metabolism and related inflammation are pertinent to a number of parkinsonian non-motor symptoms, such as pure apathy, sleep disorders, and probable rapid eye movement sleep behavior disorder (PRBD) [15,16,17]. Manganese chloride-treated zebrafish had increased pro-inflammatory cytokine levels, which may lead to neuronal degeneration and exhibit PD-like symptoms [18]. In addition, experiments conducted on animals have illustrated that brain inflammation caused by head trauma may give rise to PD-related pathology [19,20]. The bacterial endotoxin lipopolysaccharide (LPS) has the potential to induce PD through inflammatory factor release [7]. Interestingly, the initiation and progression of PD and its molecular mechanisms, pathogenicity, and symptom development present many similarities with the severe acute respiratory syndrome coronavirus 2 (SARS-CoV-2) virus and coronavirus disease 2019 (COVID-19) [21]. It has been reported that oral delivery of particular microbial metabolites to germ-free mice triggers neuroinflammation and motor symptoms. Moreover, when α-synuclein-overexpressing mice were colonized with microbiota from patients with PD, physical disabilities were exacerbated significantly [22].

Evidence suggests that the acute short-term activation of immune cells could help to protect the nervous system, leading to the repair of damaged tissue and removal of toxins and pathogens [23,24]. Acute inflammation is usually seen as advantageous, while chronic inflammation is widely believed to be associated with neurodegenerative disorders [25,26]. Injury severity and lingering immune reactions can interfere with the neurochemical processes of the central nervous system (CNS). This may cause an escalation of neuronal death, deficits in cellular maintenance capacity, a breakdown of the blood–brain barrier (BBB), and, ultimately, an overactive inflammatory response [27]. Indeed, the brain’s inflammatory response system is analogous to that of the immune system in peripheral regions, with similar characteristics and mediators [28]. The initial discovery of PD-like symptoms in individuals with encephalitis lethargica has indicated that inflammation may play a critical role in the development of parkinsonian symptoms [29]. In addition, the occurrence of activated microglia and infiltrated lymphocytes in degenerated areas of postmortem PD brains, as well as in rodent models, verifies the crucial role of inflammation in PD pathogenesis [30,31,32,33]. Research on pathological and rodent models of PD has unveiled the presence of an inflammatory response characterized by the activation of microglia, which release cytokines, and an increase of autoreactive T lymphocytes in the peripheral system. Additionally, both the peripheral nervous system and CNS have a large number of antigen-presenting cells and major histocompatibility complex (MHC) II complexes [34]. Specific molecular signatures associated with PD’s motor complications (MC) have potential utility for the early identification of subjects at risk of MC development. For example, transcription factor mRNA levels in CD4^+^ T lymphocytes in patients with PD with (PMC) and without (WMC) motor complications were assayed. The results revealed that patients WMC had increased *STAT1* and *NR4A2* expression, while patients PMC showed a higher expression of *STAT6* [35]. Interestingly, patients with PD may exhibit cognitive deficits. Further exploration demonstrated that patients with cognitive impairment displayed an elevated number of circulating lymphocytes compared to those with normal cognitive function [36]. In fact, non-motor symptoms like depression, cognitive decline, and fatigue have been found to be positively correlated with increased cerebrospinal fluid levels of inflammatory markers in PD [37]. Therefore, the pathogenesis and progression of PD are believed to be heavily influenced by both immunological activation and inflammatory processes.

The CNS requires neuroinflammation to support neuronal health. However, neurotoxicity induced by inflammation can exacerbate neuronal injury [38,39]. Herein, we review inflammatory mechanisms and therapeutic applications in PD. The summarized information may clarify the role of neuroinflammation in PD and provide directions for future research.

## 2. Inflammatory Responses and Mechanisms

### 2.1. Genes Linked to PD and Inflammatory Reaction

Many genetic locations linked to PD have been identified. Among these, we delineated genes that showed significant association with the inflammatory response (Table 1).

#### 2.1.1. LRRK2

Mutations in leucine-rich repeat kinase 2 (*LRRK2*) could lead to autosomal-dominantly inherited PD [75]. A relationship has been observed between inflammation and a particular manifestation of PD associated with the LRRK2 mutation, which is characterized by a broad spectrum of motor and non-motor symptoms [40]. In Kozina et al.’s study, using a chimeric mouse model, they present the proof that changes in the adaptive immune system of animals with *LRRK2* mutations significantly influence the formation of the neuropathological phenotype [41]. Diminishing excessive peripheral interleukin-6 (IL-6) levels prevents the decline of DA neurons in LPS-treated mutant *LRRK2* mice, suggesting that peripheral IL-6 is an essential factor in the progression of events that underlie LRRK2-mediated neuronal loss [41]. The most common mutation in PD is the glycine to serine substitution (G2019S) located in the protein kinase domain of the LRRK2 protein. In order to simulate PD pathology in vitro, Chen et al. employed 1-methyl-4-phenylpyridium (MPP^+^) to damage DA neurons and discovered that DA neurons differentiated from individuals with a G2019S *LRRK2* mutation drastically decreased the survival rate and amplified apoptosis in comparison to the controls [42]. In addition, after being exposed to MPP^+^, inflammatory factors such as *IL-1β*, tumor necrosis factor (*TNF*)*-α*, *IL-6*, cyclooxygenase-2 (*COX-2*), and inducible nitric oxide synthase (*iNOS*) with a G2019S *LRRK2* mutation showed an increase compared to the control group [42]. Interestingly, myeloid cells altered by an *LRRK2* mutation demonstrated an elevated phosphorylation and Wiskott–Aldrich syndrome protein family verprolin-homologous protein 2 (*WAVE2*) stabilization, which lead to a quicker phagocytic response [43]. Over time, the enhanced activity of microglia in the immune system could lead to a higher level of DA depletion [43].

#### 2.1.2. SNCA

From a pathological perspective, PD is identified by the presence of intraneuronal proteinaceous cytoplasmic inclusions, named Lewy bodies and dystrophic Lewy neurites, both of which include α-synuclein [76]. Consistently, the gene encoding an α-synuclein (*SNCA*) mutation was identified in familial PD [77].Fibrillogenic monomers of α-synuclein could form oligomeric intermediates, which accumulate into fibrils and eventually deposit in Lewy bodies [78]. Oligomers formed from α-synuclein monomers can activate microglia, which leads to neuroinflammation and the consequent PD progression [44]. The recruitment of microglia to the region of α-synuclein aggregation implies that α-synuclein could be a trigger for an inflammatory reaction [44]. Upon exposure of primary human microglia to α-synuclein fibrils, the inflammasome assembly process was activated, resulting in a noteworthy IL-1β secretion. Caspase-1 protein was identified to be attached to the inflammasome scaffolds [45].

In addition, mutant forms of α-synuclein (A30P and A53T) caused microglial activation, with a consequent elevation of pro-inflammatory cytokines (IL-6, IL-1β, and TNF-α) and chemokines [46,47,48]. It is noteworthy that Bido et al. created a novel mouse model through lentiviral-mediated selective α-synuclein accumulation in microglial cells. As a result, it was observed that the microglial cells with α-synuclein accumulation displayed an exhaustion of phagocytic activity and a heightened expression of pro-inflammatory molecules, and the corresponding mice exhibited DA neuron loss [49]. Microglia activated by α-synuclein could secrete matrix metalloproteinases (MMPs), which then activate protease-activated receptor-1 (PAR-1). This process consequently elicits microglial inflammatory reactions in either an autocrine or paracrine fashion [50]. The interaction between α-synuclein and toll-like receptor 4 (TLR4) caused a signaling cascade that led to the pro-inflammatory substance release. However, this response was counteracted by the expression of nuclear receptor-related 1 protein (Nurr1) [51].

#### 2.1.3. Pink1/Parkin

Mutations in *parkin*, an E3 ubiquitin ligase, and PTEN-induced kinase 1 (*Pink1*), a ubiquitin kinase, could induce early-onset PD [52]. The innate immune system can be triggered by the release of damage-associated molecular patterns (DAMPs), and mitophagy may serve to mitigate inflammation. Consistently, Sliter et al. have demonstrated that Pink1 and Parkin-mediated mitophagy could restrain innate immunity [52]. When Parkin or Pink1 is absent in mice with either acute (triggered by strenuous exercise) or chronic (caused by mtDNA mutation) mitochondrial stress, there is a pronounced inflammatory response [52]. Parkin deficiency promotes survival of activated microglia and renders DA neurons more susceptible to inflammation [53,54]. The elimination of the stimulator of interferon genes (*STINGs*), a crucial modulator of the type I interferon response to cytosolic DNA, can restore DA neuron survival and improve motor function in Parkin-deficient mice. Another effective intervention is the mating of Parkin-deficient mice with STING-null mice [52]. Similarly, Borsche et al. revealed that the circulating cell-free mitochondrial DNA (ccf-mtDNA)-cyclic guanosine monophosphate (GMP)-adenosine monophosphate (AMP) synthase (cGAS)-STING-IL-6 pathway may be responsible for Parkin/Pink1-deficiency-induced inflammation in PD [52,55]. In Pink1^−/−^ mice, intestinal infection with Gram-negative bacteria elicits mitochondrial antigen presentation and autoimmune processes, leading to the establishment of mitochondria-specific CD8^+^ T cells in the periphery and the brain [56]. Wasner et al. established induced pluripotent stem cell (iPSC)-derived neuron–microglia co-cultures from controls and patients with *Parkin* mutations, and then treated them with mtDNA isolated from Parkin-mutant cells. Their investigation revealed that, with LPS priming followed by mtDNA, *IL-6* was overexpressed in parkin-deficient cells, therefore demonstrating that parkin absences amplify cells’ response to pro-inflammatory factors [57]. Pink1-deficient mice displayed higher mRNA and protein expressions of cytokines (TNF-α, IL-1β, and IL-6) when compared to the control group [58,59]. Pink1 has been shown to directly interact with two components of the downstream signaling pathway mediated by IL-1β, namely TNF receptor-associated factor 6 (TRAF6) and transforming growth factor-β (TGF-β)-activated kinase 1 (TAK1). This interaction facilitates their activation, thereby leading to an enhanced production of cytokines mediated by IL-1β [60]. Nucleotide-oligomerization domain receptor 2 (NOD2) levels were found to be elevated in astrocytes lacking parkin, leading to the activation of endoplasmic reticulum (ER) stress and the inflammatory response [61].

#### 2.1.4. GBA1

The glucocerebrosidase 1 (*GBA1*) gene encodes the lysosomal enzyme glucocerebrosidase (GCase), a membrane-associated protein that is responsible for the cleavage of glucosylceramide (GlcCer) and glucosylsphingosine (GlcSph) [79]. *GBA1* mutations were linked to microglia activation in brain areas that are sensitive to Lewy bodies [62]. Likewise, mutant *GBA1* was capable of downregulating the wingless-related integration site (Wnt)/β-catenin signaling pathway, which negatively regulated the microglia activation [63,64]. In alignment with these discoveries, Chahine et al. elucidated that the onset of PD was found to be at a younger age and the risk of cognitive dysfunction was higher in patients who had PD with *GBA* mutations than in those with PD without *GBA* mutations. Further exploration has revealed that plasma concentrations of IL-8, monocyte chemotactic protein 1, and macrophage inflammatory protein 1α were significantly associated with *GBA* mutation status [65]. After being exposed to the competitive GCase inhibitor, conduritol-β-epoxide (CBE), mice displayed generalized inflammation, complement activation, and the insoluble aggregation of α-synuclein. This indicates that the dysfunction of GCase can exacerbate PD pathology by instigating an inflammatory response [66,67,68].

#### 2.1.5. DJ-1

Parkinson’s disease protein 7 (*DJ-1*) has been identified as a regulator of pro-inflammatory responses, and it lacking has been found to be a factor in the development of PD by causing astrocytic neuroinflammatory damage [69]. Research has demonstrated that increasing the expression of astrocyte DJ-1 in zebrafish larvae can protect them from trauma caused by the neurotoxin MPP^+^, which is related to PD [70]. Furthermore, through protein profiling of isolated astrocytes, researchers have uncovered that a rise in astrocytic DJ-1 expression activated a wide array of proteins related to inflammation [70]. Analogously, astrocytes that overexpressed hDJ-1 in animals exhibited resistance to rotenone-induced neurodegeneration, thereby manifesting a significant decrease in neuronal oxidative stress and suppression of microglial activation [71]. Utilizing shRNA to reduce DJ-1 expression in microglia, Trudler et al. observed an elevated response to dopamine, as indicated by an increased secretion of IL-1β and IL-6 [72]. Furthermore, reduced DJ-1 expression hastened microglial-induced neuroinflammation and cell apoptosis through the nuclear factor erythroid 2-related factor 2 (Nrf2)/thioredoxin 1 (Trx1)/nod-like receptor pyrin domain containing 3 (NLRP3) pathway [73]. Indeed, by stabilizing sex-determining region Y-box 9 (Sox9), DJ-1 causes prostaglandin D2 synthase (PTGDS), an enzyme responsible for the production of prostaglandin D2 (PGD2), to be expressed. Subsequently, PGD2 derived from astrocytes activates prostaglandin D2 receptor 2 (DP2) receptors in microglia, leading to anti-inflammatory heme oxygenase-1 (HO-1) expression through AKT activation [74].

### 2.2. Immune Signaling Pathways Associated with PD

Immune signaling pathways associated with PD have been investigated and are illustrated in Figure 1.

#### 2.2.1. Inflammasome

The NLRP3 inflammasome is a complex mainly composed of NLRP3, apoptosis-associated speck-like protein containing a caspase activation and recruitment domain (ASC) and caspase-1. By assembling these three components, they can respond to microbial infections or endogenous danger signals and promote the secretion of IL-1β [80,81,82,83]. When mitochondria are impaired, Pink1 will phosphorylate Parkin on the mitochondrial membrane, triggering its activation. This activated Parkin results in the ubiquitination of mitochondria [84]. A decline in Parkin function results in an accumulation of damaged mitochondria and a surplus of ROS production, which is a key factor in NLRP3-regulated inflammation [85]. Furthermore, Panicker et al. demonstrated that Parkin usually suppresses inflammasome priming by ubiquitinating NLRP3, a parkin substrate, and designating it for proteasomal degradation [86]. In addition, Parkin dysfunction leads to the accumulation of the Parkin-interacting element (PARIS) and zinc finger protein 746 (ZNF746), which in turn promotes the assembly of the NLRP3 complex, resulting in the formation of mitochondrial reactive oxygen species (mitoROS) [86]. The inhibition of the autophagy-activated NLRP3 inflammasome through phosphodiesterase 10A (PDE10A)–cyclic adenosine monophosphate (cAMP) signaling in microglia leads to a subsequent upregulation of downstream IL-1β and the augmentation of macrophage migration inhibitory factor or glycosylation-inhibiting factor (MIF), a pro-inflammatory cytokine [87,88]. The p38-transcription factor EB (TFEB) pathways prevent chaperone-mediated autophagy (CMA)-mediated NLRP3 degradation, thus activating microglia in PD [89]. ROS is a major indicator of neuronal damage in PD and is frequently seen in channelopathies, lysosomal distress, and toxicant exposure. It has been revealed that ROS induces an interaction between the leucine-rich repeats (LRRs) of NLRP3 and thioredoxin-interacting protein (TXNIP), which is necessary for autophagy regulation and affects PD-related synucleinopathy by hindering ATP13A2 [90]. Rotenone, a risk factor for PD, was observed to dose- and time-dependently activate neuronal inflammasomes, resulting in the maturation and secretion of cleaved IL-1β and IL-18 [91]. Further exploration has shown that impeding inflammasome activation could be achieved through inhibiting and eliminating cyclin-dependent kinase 5 (Cdk5) in the rotenone model [91]. The research conducted by Wang et al. suggests that the ATP-P2X4 receptor (P2X4R) is a key player in activating the NLRP3 inflammasome, which has a significant role in regulating DA neurodegeneration, dopamine levels, and glial cell activation [92]. The α-synuclein aggregates bind to toll-like receptors (TLR2), providing the initial stimulus for NLRP3 activation, prompting the nuclear translocation of downstream NF-κB, resulting in the production of NLRP3 and pro-IL-1β [93]. Interestingly, dopamine counteracts the K^+^-efflux-induced NLRP3 inflammasome stimulation in microglia [94]. Research has demonstrated that the NLRP3 being activated in the brain is not the only source of inflammation; NLRP3 activation in the liver can bring inflammatory cells into the brain, which will increase the extent of inflammation in the brain and worsen the destruction of dopamine-producing neurons [95].

#### 2.2.2. NF-κB Signaling

Analyses of the immune histochemical procedures conducted on patients with PD demonstrated an elevated translocation of NF-κB in the nucleus of the mesencephalic DA neurons, which in turn triggered the transcription of pro-inflammatory mediators (chemokines, cytokines) and the formation of ROS through auto-oxidation and enzymatic dopamine catabolism [96,97,98]. Subsequently, ROS may induce damage to neuronal cells due to lipid peroxidation, the oxidative alteration of DNA and proteins, thereby causing a neurodegenerative process [99]. Evidence suggests that preformed α-synuclein-fibril-activated TLR2 initiates the production of microglial pro-inflammatory molecules, which then raises the expression of α-synuclein in neurons via NF-κB activation [100]. Additionally, by upregulating Nurr1 and tyrosine hydroxylase (TH) and downregulating α-synuclein, NF-κB inhibition may reduce the manufacturing of inflammatory factors, thereby alleviating the inflammation in PD [101]. When NF-κB was stimulated by α-synuclein in BV-2 cells, it translocated to the nucleus, leading to the expression of TNF-α. Nurr1 counteracted this process by interacting with NF-κB/p65 and halting its nuclear movement. In addition, NF-κB and Nurr1 appeared to be impacted by the TLR4-mediated signal pathway [51]. Moreover, rotenone was reported to increase the risk of PD. Yuan et al. have discovered that rotenone toxicity may be caused by its activation of microglia, and this could be related to the NF-κB signal pathway [102]. Copper exposure has a high epidemiological correlation with PD and stimulates microglia to secrete inflammatory substances, leading to the pyroptosis of DA neurons, which was linked to the activation of the ROS/NF-κB pathway and the subsequent mitophagy [103]. Analogously, manganese could induce PD-like phenotypes and its toxic effects may be associated with p38 mitogen-activated protein kinases (MAPKs), apoptotic signaling cascades, and NF-κB [104,105].

#### 2.2.3. Toll-like Receptors (TLRs)

The inflammatory responses initiated by TLRs are well known, and microglia are known to express all TLR family members [106]. TLR signaling is initiated by identifying pathogen-associated molecular patterns (PAMPs) and DAMPs, which leads to priming and activation [107,108]. When TLR is triggered, microglia with an increased expression level can modulate inflammatory reactions [109]. Upon TLR activation, distinct adaptor proteins are expressed, including myeloid differentiation primary response gene 88 (MyD88) or the Toll-interleukin-1 receptor (TIR)-domain-containing adapter-inducing interferon-β (TRIF) pathway. Subsequently, the downstream inflammation-related signaling pathways, including MAPK, NF-κB, and interferon regulatory factor (IRF) signaling pathways, are activated [110,111]. Campolo et al. found that the absence of TLR4 diminishes the emergence of neuroinflammation related to PD via the regulation of NF-κB, activator protein-1 (AP-1), and inflammasome pathways [112]. This result was consistent with Shao et al.’s findings that TLR4 deficiency grants a protective effect against the MPTP/probenecid mouse model of PD [113]. The presence of TLR2/4 on microglial cells allows them to identify oligomeric α-synuclein, thereby activating microglia and bringing about neuroinflammation [114,115,116]. TLR4 has been observed to interact with α-synuclein and stimulate microglial responses, including the uptake of α-synuclein, the release of pro-inflammatory cytokines, and the promotion of oxidative stress [114]. Similarly, it has been demonstrated that TLR2 could directly bind to the fibrillary α-synuclein, thereby activating the production of TNF-α and IL-1β [117]. In addition, the oligomeric form of α-synuclein has been observed to interact with TLR1/2 heterodimers, which causes the p65 NF-κB subunit to enter the nucleus and activate microglia [118].

#### 2.2.4. TREM2 Receptors

The triggering receptor expressed on myeloid cell 2 (TREM2), a V-type immunoglobulin domain-containing transmembrane protein, can be found in mononuclear phagocytes such as microglia, macrophages, and osteoclasts [119]. TREM2 depletion worsened α-synuclein-induced inflammatory reactions in BV2 cells and increased apoptosis in SH-SY5Y cells exposed to a BV2-conditioned medium [120]. TREM2 knockout intensified DA neuron loss in mice when subjected to adeno-associated viral vectors expressing human α-synuclein (AAV-SYN). In addition, both in vitro and in vivo TREM2 deficiency triggered a shift from an anti-inflammatory to a pro-inflammatory activation of microglia [120]. Ren et al. revealed that TREM2 has the potential to impede the TRAF6/TLR4-induced activation of both MAPK and NF-κB pathways, which could account for its protective effects against neuroinflammation and DA cell destruction in the MPTP mouse model of PD [121]. Interestingly, a possible reason for the neuroinflammation and neurodegeneration observed in PD is an over-activation of M1 microglia and an insufficient M2 microglia response [122]. Based on this knowledge, the repression of TREM2 switches the protective M2 microglia to the inflammatory M1 phenotype, thus intensifying neuroinflammation in PD [122].

#### 2.2.5. MAPK Signaling

In Gao et al.’s study, they realized that rotenone directly activated the NF-κB signaling pathway in BV2 cells. The activated NF-κB signaling pathway, which was dependent on p38 MAPK, caused considerable inflammatory cytokine production and microglial activation [123]. MAPK-activated protein kinase 2 (MK-2) is activated by phosphorylated p38α, resulting in elevated levels of TNF-α, IL-1β, and IL-6 [124,125]. In addition, in order to protect against PD, it is essential to avoid the initial rise of TNF, which triggers protein thiol oxidation and activates apoptosis signal-regulating kinase 1 (ASK1)-p38 signaling [126]. LPS-induced neurotoxicity in primary mesencephalic neuron–glia co-cultures was significantly reduced in DA neurons from MK2-deficient mice, in comparison to those from wild-type mice [127]. MK2-deficient cultures showed reduced inflammation and subsequent neuroprotection, as evidenced by decreased levels of the keratinocyte-derived chemokine, TNF-α, IL-6, and NO [127].

#### 2.2.6. JAK/STAT Pathway

In vitro, the exposure of microglia and macrophages to α-synuclein triggered the activation of the janus kinase (JAK)/signal transducer and activator of transcription (STAT) pathway. However, when these cells were treated with AZD1480, a JAK1/2 inhibitor, the induction of MHC II and inflammatory genes caused by α-synuclein was prevented [128]. Consistently, researchers have found that the SN of rats with increased α-synuclein levels show a heightened expression of inflammatory and neurological disease-related genes, which decreases upon the administration of AZD1480 [128]. It has been proven that STAT3 can activate microglia cells, which are responsible for the production of cytokines such as interferon (IFN)-γ, TNF-α, IL-1β, IL-6, and IL-23, in PD [129]. Through the activation of nicotinamide adenine dinucleotide phosphate (NADPH) oxidase and iNOS, cytokines induced the production of reactive oxygen species (ROS) and nitric oxide (NO), leading to the degeneration of DA neurons [129]. Stimulating STAT1 and STAT3 activity in patients with PD causes a dramatic rise in Th1 and Th17 levels [130]. Recruitment of inflammatory cytokines such as IFN-γ, TNF-α, IL-1β, and IL-23 through the JAK/STAT activation is crucial in inducing ROS, nitric oxide species (NOS), and the decline of DA neurons [131].

#### 2.2.7. RAGE

It is acknowledged that the receptor for advanced glycation end products (RAGE) is an essential regulator of neuroinflammation [132]. RAGE has been identified as the primary cause of the pro-inflammatory effects of synuclein pre-formed fibrils (PFFs) on microglia, including the expression of TNF-α, IL-1β, and IL-6, as well as the phosphorylation of p38 MAPK [133]. Wang et al.’s results suggested that the activation of p38 MAPK triggers the secretion of pro-inflammatory cytokines, facilitated by the RAGE-NF-κB pathway. This, in turn, leads to the degradation of dopamine in the striatum and the eventual loss of DA neurons in the SN [134]. Utilizing its positively charged surface, the V domain of RAGE can connect to the acidic C terminus of α-synuclein, thus allowing the binding of α-synuclein amyloid fibrils to microglia. The effects of α-synuclein fibril-induced neuroinflammation can be inhibited by blocking RAGE [133]. Likewise, by hindering RAGE, astrocyte and microglia activation was blocked, and the injection of FPS-ZM1, a multimodal blocker of RAGE, decreased the circulating cytokines in serum and cerebrospinal fluid (CSF). Accordingly, the loss of tyrosine hydroxylase and NeuN-positive neurons was significantly hindered due to RAGE blocking [135].

#### 2.2.8. Others

By activating the Wnt/β-catenin pathway, inflammation is attenuated and neuroprotection is improved owing to multiple interactions between microglia/macrophages and astrocytes [136,137]. Rho-kinase (ROCK) is increasingly being viewed as a promising therapeutic target for inflammatory neurodegeneration diseases [138]. ROCK inhibition has been found to be beneficial in a PD model induced by intranasal LPS. It protects dopamine neurons and reduces inflammation [139]. It was discovered that glycogen synthase kinase-3β (GSK-3β) plays a major role in controlling the inflammatory response. 6-Hydroxydopamine (6-OHDA) was associated with a considerable dephosphorylation/activation of GSK-3β and nuclear translocation of NF-κB p65. LiCl and SB415286, which are inhibitors of GSK-3β, suppressed the GSK-3β/NF-κB signaling pathway, resulting in the diminution of pro-inflammatory molecules in astrocytes activated by 6-OHDA [140]. GSK-3β is an essential factor in sustaining the vicious cycle between activated microglia and damaged neurons, which is the cause of DA cell depletion in PD [141]. Ma et al. observed that the absence of cGAS in microglia effectively regulated the inflammatory response and neurotoxic effects triggered by MPTP [142]. From a mechanical perspective, the removal of cGAS lessened neuronal impairment and inflammation in astrocytes and microglia by hindering the activation of antiviral inflammatory pathways [142]. In addition, it has been put forward that NO may be involved in the inflammatory responses seen in PD [143]. GW274150, a selective iNOS inhibitor, offers neuroprotection and reduces iNOS expression as well as the inflammatory reaction in the 6-OHDA-induced PD model [144].

## 3. Anti-Inflammatory Therapies for PD

There are many treatment methods for PD. In this section, we provide a summary of these inflammation-related therapies according to their mode of action.

### 3.1. NLRP3 Inflammasome Inhibition

The Antrodia camphorata polysaccharide (ACP) is the main component of the natural polyporaceae aphididae. In MES23.5 cells, ACP hindered the expression of ROS-NLRP3 induced by 6-OHDA, providing a protective effect, and decreased the activation of ROS-NLRP3 in the SN–striatum, thereby augmenting the exercise capacity of PD mice [145]. In addition, ACP led to an upregulation of dopamine in the striatum of 6-OHDA-induced PD mice, along with a noticeable reduction in NLRP3 inflammasome signaling in the same region [146]. Through the inhibition of NLRP3 inflammasome activation, the small molecule kaempferol was able to provide protection to mice from LPS and α-synuclein-induced neurodegeneration [147]. In addition, kaempferol reduces iNOS, COX-2, and IL-18 formation by preventing NF-κB, p38 phosphorylation and NLRP3 inflammasome activation in the SN of PD rats caused by 6-OHDA [148]. By means of an NLRP3 inhibitor, MCC950, microglial inflammasome activation was successfully inhibited, DA neurons in the SN were preserved, and motor deficits were ameliorated [149]. Rhynchophylline was found to be effective in alleviating the behavioral deficits caused by MPTP, reducing DA neuron destruction, and reversing inflammatory cytokine and oxidative stress indicator production, which were attributed to TLR4/NLRP3 pathway modulation [150]. Similarly, genkwanin managed to restrain the MPP^+^-induced TLR4/MyD88/NLRP3 inflammasome pathway activation in SH-SY5Y cells [151]. Andrographolide could rescue DA neuron loss and was found to be effective in promoting the formation of a parkin-dependent autophagic flux in microglia, which facilitated the clearance of impaired mitochondria and suppressed NLRP3 inflammasome activation [152]. The phosphorylation levels of the phosphatidylinositol 3-kinase (PI3K)/protein kinase B (Akt)/GSK3β signaling pathway, ROS production, and p65 phosphorylation were notably reduced by fingolimod, a sphingosine-1-phosphate receptor antagonist, which inhibited NLRP3 inflammatory bodies, thus indicating that fingolimod could be a potential therapeutic approach for PD [153]. The utilization of icaritin and its glucoside in PD mice induced by MPTP or 6-OHDA lessens DA neuronal harm, reduces the secretion of pro-inflammatory cytokines, and inhibits the protein expression of ionized calcium-binding adapter molecule 1 (Iba-1) and glial fibrillary acidic protein (GFAP) in the brain through impeding NLRP3 inflammasome activation and spurring the production of kelch-like ECH-associated protein 1 (Keap1), HO-1, Nrf2, and NADPH quinone dehydrogenase 1 (NQO1) proteins [154,155]. The process is shown in Figure 2.

### 3.2. NF-κB Inhibition

The compound known as hypoestoxide has been recognized for its ability to affect the activity of NF-κB. In a study involving transgenic mice that overexpressed a-synuclein in their mThy1 cells, the intraperitoneal administration of hypoestoxide (5 mg/kg) on a daily basis for a period of four weeks yielded notable results. These included a reduction in the release of pro-inflammatory cytokines from microglia, an amelioration of the loss of DA neurons, as well as improvements in motor behavioral deficits. Additionally, hypoestoxide exhibited a significant decrease in the levels of nuclear phosphorylated NF-kB within the frontal cortex of a mouse model of PD [156]. Similarly, PD180970, a chemical biology-synthesized small molecule, manifests an anti-neuroinflammatory effect by inhibiting the secretion of pro-inflammatory cytokines, specifically IL-6 and monocyte chemoattractant protein-1, via the downregulation of NF-κB activation [157]. Research carried out on MPTP-intoxicated mice affirmed that chlorogenic acid, through its action of downregulating the NF-κB-mediated neuroinflammatory pathway, effectively safeguards against DA neuron degeneration in the SN, a hallmark characteristic of PD [158]. It has been ascertained that polyphenols can act as anti-inflammatory agents, thereby interrupting the oxidative injury through the suppression of NF-κB [159]. The CU-CPT22, a small-molecule compound, exhibits inhibitory effects on the heterodimer TLR1/2. It possesses the capability to diminish the nuclear translocation of NF-κB, as well as suppress the secretion of TNF-α, in cultured primary mouse microglia via an MyD88-dependent mechanism [118]. When exposed to the neurotoxic MPP^+^, morin, a flavonoid found in *Maclura pomifera*, *Maclura tinctoria*, and the leaves of *Psidium guajava*, may diminish cell damage through inhibiting astroglial activation and the nuclear translocation of NF-κB [160,161]. Another flavonoid, namely, baicalein, derived from the roots of the Chinese medicinal herb *Scutellaria baicalensis*, exhibited a mitigating effect on motor impairment and astrocyte activation in rodent models of PD induced by MPTP. This remarkable improvement was achieved by inhibiting the production of inflammatory cytokines and the activation of NF-κB [162]. Calycosin, an isoflavone phytoestrogen that is isolated from *Astragalus membranaceus*, could ameliorate the behavioral abnormalities, reduce pro-inflammatory molecule levels, and maintain TH neuron integrity in the brain of MPTP-induced PD mice by suppressing the expression of TLR2, TLR4, the MAPK pathway, and nuclear NF-kB [163]. Tanshinone-I and α-asarone possess the capacity to selectively impede the NF-κB pathway. In murine models of PD, the use of these two drugs inhibits the differentiation of the M1 subtype and reduces the levels of pro-inflammatory components [164,165]. Vildagliptin, a selective antagonist of dipeptidyl peptidase-4, could mitigate the heightened expression of RAGE and its downstream mediator NF-κB triggered by rotenone, thereby ameliorating the synthesis of inflammatory regulators such as myeloperoxidase, TNF-α, and iNOS [166]. Lenalidomide was given to mThy1-α-synuclein transgenic animals, which resulted in a reduction in the expression of pro-inflammatory cytokines TNF-α, IL-6, IL-1β, and IFN-γ, an increase in the expression of the anti-inflammatory cytokines IL-10 and IL-13, and an inhibition of NF-κB signaling [167]. The process is shown in Figure 3.

### 3.3. Microglia Inhibition

Following JWH133 (selective cannabinoid receptor agonist) administration in MPTP-induced PD animal models, there is a decrease in the levels of pro-inflammatory factors produced by microglia, as well as a reduction in the migration of peripheral immune cells to the CNS [168]. 1, 25-dihydroxyvitamin D3 is the active metabolite of vitamin D3 and confers protective effects in a rodent model of PD via inhibiting microglial activation [169]. α-Lipoic acid, a water- and oil-soluble antioxidant found in the mitochondria, could lead to a decline in microglial activation in the SN, a decrease in inflammatory factors, and an enhancement of motor capability in the MPTP mice model of PD [170]. Montelukast, a cysteine leukotriene receptor antagonist, was found to rescue DA neurons from degeneration via inhibiting microglial activation, reducing p38 MAPK and p53, and decreasing the production of neurotoxic cytokines including TNF-α and IL-1β in a mouse model of PD [171,172]. Evidence suggests that acacetin has the potential to regulate NO, prostaglandin E2, and TNF-α in vitro, and can also hinder microglial activation, iNOS, and COX-2 expression, effectively mitigating neuroinflammation and locomotor deficits in an MPTP mouse model [173]. Research has revealed that the reduction in damage to DA neurons, the inhibition of microglia activation, and the improvement in animal behavior can be achieved by injecting JNJ7777120, a histamine receptor 4 (H_4_R) antagonist, into the lateral ventricle [174]. It has been suggested that minocycline, a derivative of tetracycline that can cross the blood–brain barrier, has the potential to enhance neuronal endurance in PD. This molecule has been portrayed as an inhibitor of the induction and propagation of microglia and the release of pro-inflammatory cytokines from activated microglia [175,176]. Artemisinin’s potential to reduce damage to DA neurons in a PD mouse model may be achievable by decreasing microglial activation through the TLR4-mediated MyD88-dependent signaling pathway [177]. Triptolide, an innate anti-inflammatory constituent extracted from the Chinese botanical *Tripterygium wilfordii Hook F*, enhances the expression of metabotropic glutamate receptor 5 and subsequently suppresses the activation of microglia in the LPS-induced model of PD [178,179]. An MPTP-induced PD mouse model showed that the co-treatment of ascorbic acid and NXP031, a novel single-stranded DNA aptamer that heightens the efficacy of ascorbic acid by connecting to it and reducing its oxidation, could mitigate DA neuron destruction and neuroinflammation caused by microglia in the SN [180]. Naringin defends DA neurons by curbing microglial activation and enhancing the mTOR pro-survival pathway [181]. Capsaicin induces a shift of pro-inflammatory microglia to an anti-inflammatory state, thus preventing the degeneration of SN DA neurons and the expression of pro-inflammatory markers in an LPS mouse model of PD [182]. Ginsenoside Rg1 administration diminished microglial activation and the influx of CD3^+^ T cells into the SN of MPTP-lesioned animals [183]. The vasoactive intestinal peptide (VIP) is a natural hormone that fosters neuroprotection through augmenting regulatory T cell population and enhancing their efficacy. It also inhibits microglial activation and alleviates neuronal degeneration [184,185,186]. Interestingly, the VIP receptor 2 agonist, LBT3627, could mitigate microglial reactions and enhance neuronal preservation by fostering an anti-inflammatory microenvironment, consequently modulating Th1/Th17 cytokine reactions in both the MPTP-induced mouse model and the α-synuclein overexpression model in rats [186,187,188,189]. The process is shown in Figure 4.

### 3.4. Astrocyte Inhibition

Silibinin, a prominent constituent of silymarin, could mitigate motor impairment, neuronal degeneration, and astrocyte activation in a mouse model of PD induced by MPTP through inhibiting extracellular signal-regulated kinase (ERK) and Jun N-terminal kinase (JNK) signaling pathways and COX expression [190]. Wang et al. observed that the takeda G protein-coupled receptor 5 (TGR5) inhibitor triamterene could reduce the anti-neuroinflammatory effect of bear bile powder on LPS-stimulated rat C6 astrocytic cells, suggesting that bear bile powder’s protective effects on PD mice may threaten astrocyte-mediated inflammation via TGR5 [191]. The process is shown in Figure 5.

### 3.5. NADPH Oxidase Inhibition

The NADPH oxidases play a crucial role as membrane-associated, multi-component enzymatic assemblies that facilitate the transfer of electrons across the plasma membrane, from NADPH to molecular oxygen, and the production of superoxide free radicals [192]. Diphenyleneiodonium (DPI), which is a specific inhibitor of NADPH oxidase, can mitigate neuroinflammation through the inhibition of the NADPH pathway, thereby protecting DA neurons [193]. In mouse models of PD induced by paraquat and maneb, taurine has been found to significantly decrease DA neurodegeneration and α-synuclein oligomerization by inhibiting the microglial M1 phenotype through blocking NADPH oxidase activation [194]. The process is shown in Figure 5.

### 3.6. PPARγ Agonist

In a mouse model of PD, MHY908, a dual agonist of PPARα/γ, was noted to provide protection against MPTP-induced DA neuron loss through its ability to diminish MPTP-induced neuroinflammation and suppress the activation of microglia, as reported in [195]. As a PPARγ agonist, rosiglitazone’s protective effects against PD can be ascribed to its ability to modify microglial polarization through the upregulation of anti-inflammatory cytokines associated with the M2 phenotype, while concurrently downregulating pro-inflammatory cytokines associated with the M1 phenotype [196]. It has been shown that PPARγ agonist pioglitazone can diminish the inflammatory reaction, maintain DA nigrostriatal performance, and improve PD manifestations [197,198]. The process is shown in Figure 5.

### 3.7. Targeting MAPK Pathway

In PD, the pharmacological inhibition of p38 MAPK (e.g., doxycycline) endowed neuroprotective effects [199,200]. An in vivo study revealed that when given via an intranigral LPS injection, vanillin had a marked effect in decreasing inflammatory responses, enhancing motor performance, curbing DA neuron degeneration [201]. Further exploration illuminated that the protective effect of vanillin may depend on its ability to regulate ERK1/2 and p38 MAPK signaling [201]. Interestingly, SKF, a compound that attenuates inflammation by inhibiting p38α/β, could lead to a decrease in the levels of TNF, IL-6, and IL-13 in the murine model of PD, suggesting that the MAPK pathway may occupy a vital role in regulating PD [202]. The process is shown in Figure 5.

### 3.8. Targeting AMPK-Dependent Pathway

Adenosine monophosphate-activated protein kinase (AMPK)-induced HO-1 expression in microglia, stimulated by KMS99220, a morpholine-containing chalcone compound, is a major contributor to the early anti-inflammatory response [203]. Cao et al.’s investigation was implemented and the results revealed that liraglutide confers protective effects against MPTP-caused damage in mice via AMPK/NF-κB signaling pathway-mediated neuroinflammation [204]. Dexmedetomidine has the capacity to reduce the dorsal horn of the spinal cord astrocyte activation and inflammation by activating AMPK and blocking the mammalian target of the rapamycin (mTOR)/NF-κB pathway, therefore protecting DA neurons in PD [205]. It was reported that indole-3-carbinol showed neuroprotective effects against rotenone-induced PD by activating the silent-mating-type information regulation 2 homolog 1 (SIRT1)-AMPK signaling pathway and inhibiting the subsequent inflammatory process [206]. Interestingly, mice subjected to MPTP treatment and later receiving fecal microbiota transplantation from patients with PD exhibited a pronounced deterioration in motor capabilities, degeneration of DA neurons, heightened activation of glial cells within the nigrostriatal tract, inflammation in the colon, and disturbance in the AMPK/superoxide dismutase2 (SOD2) signaling pathway [207]. The process is shown in Figure 5.

### 3.9. Targeting α-Synuclein

Administering anti-TLR2 to a high-α-synuclein-expressing mouse was found to alleviate α-synuclein aggregation in neural and astrocytic cells, neuroinflammation, neurodegeneration, and behavioral impairments [208]. A variety of clinical trials exploring the use of anti-α-synuclein antibodies have been carried out. PD01A and PD03A are two examples of active immunotherapies that work using synthetic peptides that resemble epitopes on α-synuclein to trigger a potent humoral immune response [209,210]. Through the combination of the two peptides with the carrier protein keyhole limpet hemocyanin and the adjuvant of aluminum hydroxide, T cells were able to identify the epitope and stimulate the production of antigen-specific antibodies from plasma cells [209]. The antigenic peptides were formulated with the intention of alleviating α-synuclein immune tolerance through eliciting B-cell mobilization and generating elevated levels of antibodies toward the immunizing peptide, while avoiding the initiation of an autoimmune reaction [209,210]. In experiments with rat brain models, baicalein was established to inhibit α-synuclein aggregation, inflammasome activation, and cathepsin B production during MPP^+^ infusion [211]. The process is shown in Figure 6.

### 3.10. Targeting miRNAs

The conserved gene microRNA-7 (*miR-7*) is acknowledged for its ability to reduce α-synuclein and defense against oxidative stress [212]. An analysis demonstrated that *miR-7* protected the DA neurons and abated the neuroinflammation from exogenous α-synuclein preformed fibrils’ toxicity in a mice model of PD [213]. The administration of *miR-7* mimetics through stereotactic injection into the striatum of mice can prevent the degeneration of DA neurons and alleviate microglial activation in mouse models of PD [214]. In addition, Zhou et al. unraveled that NLRP3 is a target gene of *miR-7* and the application of *miR-7* mimics inhibits NLRP3 inflammasome activation, thereby safeguarding DA neurons from degeneration in a mouse model of PD [214]. Lv et al. exposed mice to MPTP and rosmarinic acid. Their results revealed that rosmarinic acid reduces inflammation, apoptosis, and oxidative stress by controlling *miR-155-5p* [215]. *miR-3473b* could be responsible for the regulation of pro-inflammatory mediators by targeting TREM2/UNC51-like kinase-1 (ULK1) expression to control the role of autophagy in the cause of inflammation in PD, implying that *mir-3473b* could be a feasible therapeutic target to manage the inflammatory response in PD [216]. The process is shown in Figure 6.

### 3.11. Acupuncture

Studies in MPTP-induced PD models have revealed that acupuncture at Yanglingquan (GB34) and Taichong (LR3) can reduce the expression of macrophage antigen complex-1 (MAC-1), an indicator of microglial activation, and also prevent the upsurge of COX-2 and iNOS expression [217]. In addition, acupuncture suppresses inflammatory responses and apoptosis as it increases the levels of DA fibers and neurons in the SN and striatum; lowers GFAP, Iba-1, NF-κB, and TNF-α; and re-establishes the conversion of Bax and Bcl-2 expression in the SN and striatum [218]. The process is shown in Figure 6.

### 3.12. Exercise

Physical exercise could reduce α-synuclein, alleviate neuroinflammation, promote synaptic connectivity, and prolong DA neuron survival in PD [219,220,221,222,223]. Exercise on a treadmill reduces oxidative stress and facilitates mitochondrial synthesis. As a consequence, it diminishes inflammation, boosts the clearance of α-synuclein, and promotes motor symptoms among individuals with PD [220,223]. Analogously, Sung et al. and Al-Jarrah et al. have also posited that exercise may diminish the inflammatory response in the experimental model of PD [224,225]. Tyrosine hydroxylase (TH) is essential for DA generation as it is the rate-limiting enzyme in the catecholaminergic biosynthetic pathway [226]. It was indicated that prolonged exercise could reduce pro-inflammatory cytokines TNF-α and IL-1β, enhance antioxidant capacity, and restore the levels of TH [222,227]. Endurance training exerted a suppressive effect on the expression of α-synuclein and counteracted neuroinflammation induced by MPTP through impeding the downstream signaling pathways of TLR2 in the cerebral regions of MPTP-intoxicated mice [222]. The process is shown in Figure 6.

### 3.13. Others

Patients with inflammatory bowel disease (IBD) exhibited a greater prevalence of PD compared to individuals unaffected by IBD. The administration of anti-inflammatory anti-tumor necrosis factor (anti-TNF) therapy at an earlier stage correlated notably with a significant decrease in the occurrence of PD [228]. Exendin is an analog of glucagon-like peptide-1 (GLP-1) that specifically attaches itself to the GLP-1 receptor [229]. An investigation using exendin highlighted that exendin might confer neuroprotective effects via attenuating matrix metalloproteinase-3 (MMP-3) upregulation, decreasing pro-inflammatory cytokine expression, and protecting against DA neuron destruction [230]. Neuroinflammation in PD has been attributed to COX-2. Reduced COX-2 activity has been observed in MPTP-induced PD mice following melatonin administration [231,232]. Cyclosporin improved motor performance and showed an anti-inflammatory effect through reducing the nuclear factor of activated T cells, cytoplasmic 3 (NFATc3), and diminishing mitochondrial stress in the midbrain of MPTP-injured animals [233]. Dietary strategies that target the relationship between the gut and the brain have the potential to retard PD progression and relieve symptoms such as gastrointestinal dysfunction by altering inflammatory pathways [234]. For example, probiotics have been found to be effective in reducing inflammation and lessening PD-related gastrointestinal and behavioral symptoms such as abdominal pain, bloating, bradykinesia, rigidity, tremors, and gait dysfunction, suggesting a vital role of the inflammatory response [235,236].

As reported by epidemiological studies, those using nonsteroidal anti-inflammatory drugs (NSAIDs) chronically have a lower rate of idiopathic PD than their age-matched counterparts who do not use these medications [237]. Ibuprofen, by inhibiting COX and thereby reducing COX-induced ROS generation, lessens the inflammation, prevents the DA decrease caused by MPTP, and endows partial protection against MPTP toxicity [238]. In a preliminary study, using indomethacin before the MPTP presented a drop in microglia and lymphocyte infiltration as well as a protection of DA neurons [239].

In addition, some extracts may also play an anti-inflammatory role, which in turn relieves PD symptoms. The preliminary administration of polymannuronic acid could offer protection against PD progression by suppressing inflammation in the gut, brain, and systemic circulation, as well as reinforcing the integrity of the intestinal barrier and blood–brain barrier [240]. In rotenone-induced PD, resveratrol has been suggested to diminish ERstress, reduce IL-1β, and augment Nrf2 DNA binding activity [241]. By taking resveratrol, antioxidant enzyme capacity is increased, and COX–2, phospholipase A2 (PLA2), and TNF–α are concomitantly decreased, thus leading to enhanced mitochondrial function and suppressed neuroinflammation in 6-OHDA rodent PD models [242]. Mice treated with curcuminoids showed resistance to MPTP-induced loss of TH-positive neurons and the reduction of dopamine levels in the striatum, along with a decrease in cytokines, total nitrite, and inducible nitric oxide synthase [243]. In rodent PD models, eigallocatechin-3-gallate (EGCG) has been observed to heighten TH activity while simultaneously reducing iNOS, TNF–α, IL–6, and nitrite [242,244]. The supplementation of carvacrol orally in rats injected with 6-OHDA markedly prevents the decline of DA neurons and reduces the levels of pro-inflammatory cytokines [245]. The process is shown in Figure 7. Considering the complexity and diversity of these potential anti-inflammatory treatments for PD, we summarized these anti-inflammatory strategies in Table 2.

## 4. Discussion

Currently, the primary approach for treating symptoms is to replace dopamine using levodopa (L-DOPA), which is considered the most effective; however, this method of treatment brings about undesirable long-term effects such as dyskinesia and involuntary abnormal movements, thus restraining its use to manage symptoms [253,254]. Once more intense symptoms start to appear during the later phases of the disease, treatment usually begins. Nevertheless, this is when considerable and irreversible neural degeneration has already occurred [255,256]. Therefore, it is urgent to find a more effective and safer therapeutic strategy. Inflammation is an early event in PD progression and plays a vital role in PD pathogenesis. Therefore, several clinical studies have been conducted to determine drugs that can potentially treat PD via modulating the inflammatory response. For example, the efficacy of DNL151, a powerful LRRK2 inhibitor, is being evaluated in clinical trials for its ability to slow down the progression of pathology in patients with early-stage PD [249]. Drugs targeting α-synuclein, such as PD01A and PRX002, were tested in clinical trials and the results have showed that they may serve as promising candidates for PD management [210,247]. Selnoflast, an NLRP3 inflammasome inhibitor, was evaluated in a phase 1 clinical trial for early idiopathic PD [246]. Gendelman et al. conducted a placebo-controlled double-blind phase 1 trial and found that sargramostim, a granulocyte–macrophage colony-stimulating factor (GM-CSF), could improve PD in comparison to the placebo [250]. The positive results of clinical studies on exenatide, a GLP-1R agonist, have sparked interest in its potential for treating PD. The noteworthy effects have prompted the launch of a phase 3 clinical trial to evaluate exenatide as a potential drug [251]. By taking rifaximin, a non-absorbable antibiotic, the gut microbiota can be modified. Patients with PD and higher baseline inflammation levels may have experienced benefits from taking rifaximin [252]. The negative association between ibuprofen and PD risks implies that ibuprofen has the potential to act as a neuroprotective agent against PD (see Table 2) [248].

Herein, we review the genes associated with PD, including *LRRK2*, *SNCA*, *Pink1/Parkin*, *GBA1*, and *DJ-1*. In addition, the inflammatory mechanism related to PD may be through modulating inflammasomes, NF-κB signaling, TLRs, TREM2 receptors, MAPK signaling, JAK/STAT, RAGE, Wnt/β-catenin, ROCK, GSK-3β, and cGAS. Based on these potential mechanisms underlying PD, various anti-inflammatory therapies (i.e., NLRP3 inflammasome inhibition, NF-κB inhibition, microglia inhibition, astrocyte inhibition, NADPH oxidase inhibition, PPARγ agonist, targeting MAPK pathway, targeting AMPK-dependent pathway, targeting α-synuclein, targeting miRNAs, acupuncture, exercise, NSAIDs, etc.) were explored to alleviate the disease. Despite being arranged hierarchically in this review, mechanisms and therapeutic drugs are intertwined. Some drugs may exert anti-parkinsonian effects through multiple mechanisms (e.g., inhibition of microglial activation, pro-inflammatory molecule generation, NF-κB signaling, and NLRP3 inflammasome activation). Therefore, these inflammatory mechanisms and therapies cannot occur in isolation and numerous interactions between inflammatory pathways exist, making investigations into the role of inflammation in the development of PD challenging.

Our review offers a summary of the inflammatory effects, mechanisms, and therapeutic strategies of PD. Although a lot of work has been done in this area, much research is still in the basic science and development stage. Special attention should be paid to drug development, therapeutic applications, and clinical research. Meanwhile, understanding the potential mechanisms and strategies may facilitate future PD treatment.

## Figures and Tables

**Figure 1 ijms-25-01641-f001:**
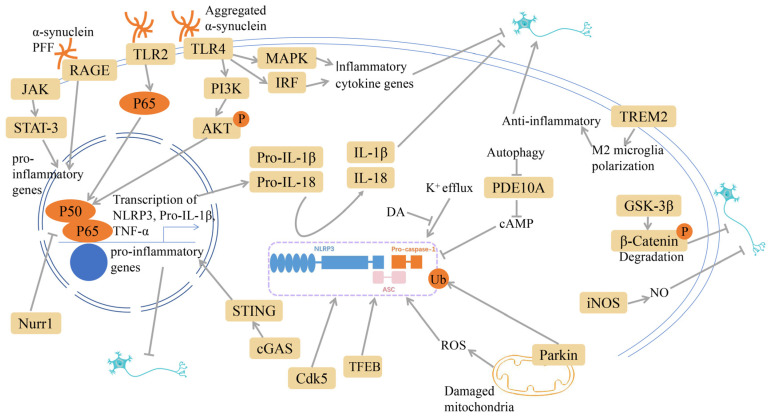
Immune signaling pathways associated with Parkinson’s disease (PD). Modulating nod-like receptor pyrin domain containing 3 (NLRP3) inflammasome, nuclear factor κB (NF-κB) signaling, toll-like receptors (TLRs), triggering receptor expressed on myeloid cell 2 (TREM2) receptors, mitogen-activated protein kinase (MAPK) signaling, janus kinase (JAK) /signal transducers and activators of transcription (STAT), receptor for advanced glycation end products (RAGE), wingless-related integration site (Wnt)/β-catenin, glycogen synthase kinase-3β (GSK-3β), and cyclic GMP-AMP synthase (cGAS) may affect the occurrence and development of PD. Grey arrows mean promotion and grey blunt-ended bars mean inhibition.

**Figure 2 ijms-25-01641-f002:**
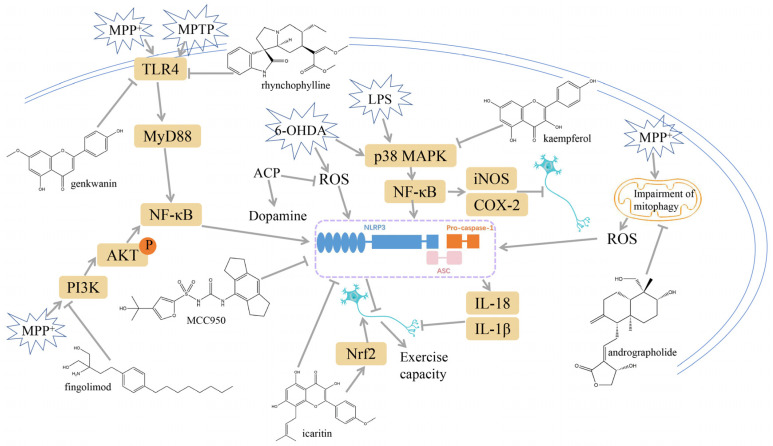
Inflammasome inhibition may offer a therapeutic benefit in the treatment of PD. NLRP3 inflammasome is a protein complex composed of apoptosis-associated speck-like protein containing a caspase recruitment domain (ASC), NLRP3 protein, and pro-caspase-1. Antrodia camphorata polysaccharide (ACP), kaempferol, MCC950, rhynchophylline, genkwanin, andrographolide, fingolimod, and icaritin may attenuate PD via NLRP3 inflammasome inhibition. The chemical structures were drawn using ChemDraw software (version 18.0.0.231). Grey arrows mean promotion and grey blunt-ended bars mean inhibition.

**Figure 3 ijms-25-01641-f003:**
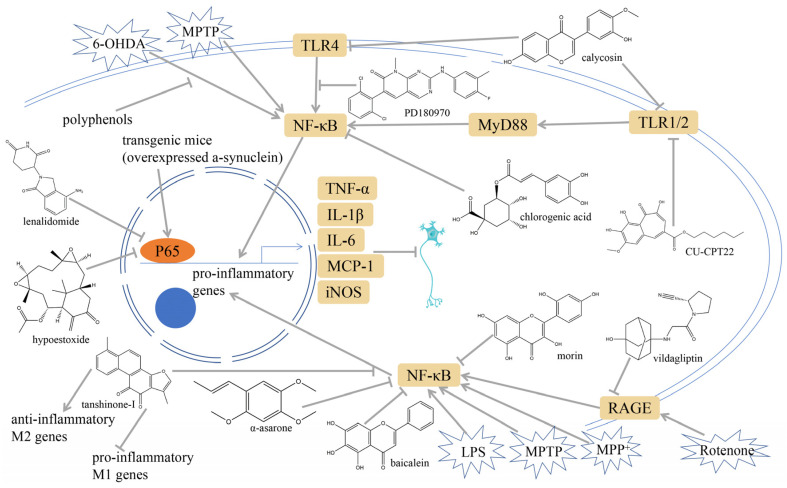
NF-κB inhibition may alleviate PD. Hypoestoxide, PD180970, chlorogenic acid, polyphenols, CU-CPT22, morin, baicalein, calycosin, tanshinone-I, α-asarone, vildagliptin, and lenalidomide confer protective effects against PD through the inactivation of NF-κB pathway. The chemical structures were drawn using ChemDraw software (version 18.0.0.231). Grey arrows mean promotion and grey blunt-ended bars mean inhibition.

**Figure 4 ijms-25-01641-f004:**
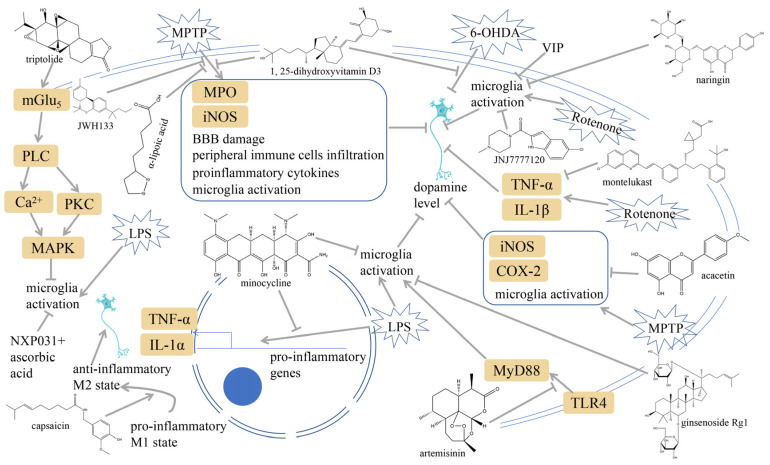
Microglia inhibition may serve as a therapeutic target in the management of PD. JWH133, 1,25-dihydroxyvitamin D3, α-lipoic acid, montelukast, acacetin, JNJ7777120, minocycline, artemisinin, triptolide, ascorbic acid and NXP031, naringin, capsaicin, ginsenoside Rg1, and vasoactive intestinal peptide could modulate PD by suppressing microglia activation. The chemical structures were drawn using ChemDraw software (version 18.0.0.231). Grey arrows mean promotion and grey blunt-ended bars mean inhibition.

**Figure 5 ijms-25-01641-f005:**
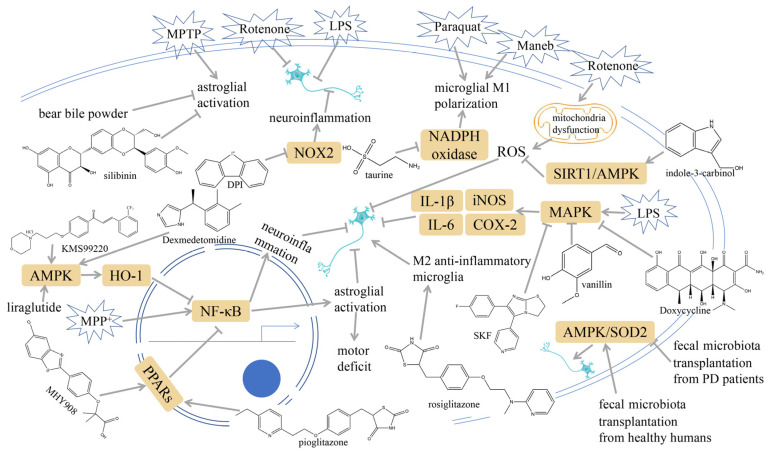
Targeting astrocyte cells, nicotinamide adenine dinucleotide phosphate (NADPH) oxidase, peroxisome proliferator-activated receptor γ (PPARγ), MAPK pathway, and adenosine monophosphate-activated protein kinase (AMPK) pathway may yield strategies to mitigate PD. The potential components or therapies are as follows: silibinin, bear bile powder, diphenyleneiodonium (DPI), taurine, MHY908, rosiglitazone, pioglitazone, doxycycline, vanillin, SKF, KMS99220, liraglutide, dexmedetomidine, indole-3-carbinol, fecal microbiota transplantation. The chemical structures were drawn using ChemDraw software (version 18.0.0.231). Grey arrows mean promotion and grey blunt-ended bars mean inhibition.

**Figure 6 ijms-25-01641-f006:**
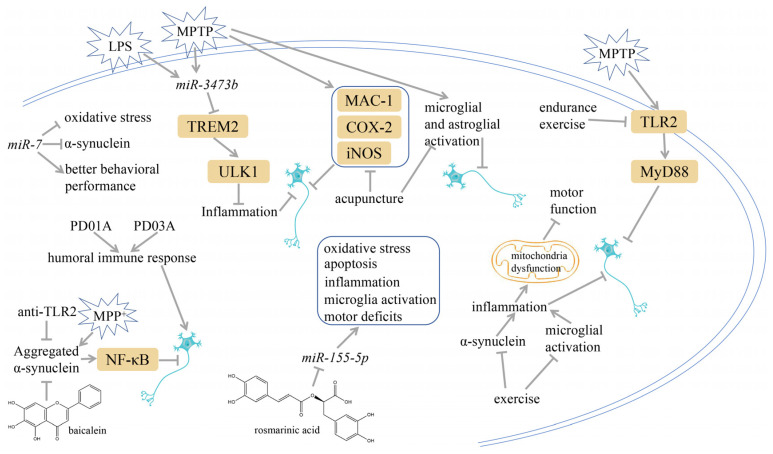
Targeting α-synuclein, modulating miRNAs, acupuncture, and exercise may attenuate PD. The potential components or therapies are as follows: anti-TLR2, PD01A, PD03A, baicalein, miR-7, rosmarinic acid, miR-155-5p, miR-3473b, acupuncture, and exercise. The chemical structures were drawn using ChemDraw software (version 18.0.0.231). Grey arrows mean promotion and grey blunt-ended bars mean inhibition.

**Figure 7 ijms-25-01641-f007:**
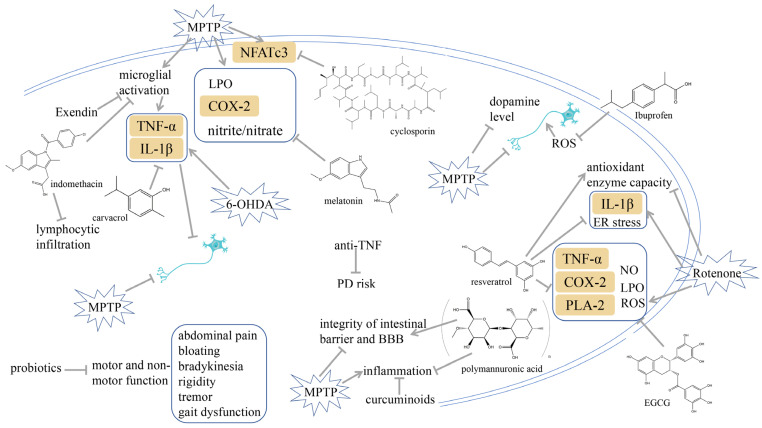
The potential anti-inflammatory treatments for PD. Reduced inflammatory response was observed in anti-tumor necrosis factor (TNF), exendin, melatonin, cyclosporin, probiotics, nonsteroidal anti-inflammatory drug (NSAID), polymannuronic acid, resveratrol, curcuminoid, eigallocatechin-3-gallate (EGCG), and carvacrol treatments, which may be the reason underlying the attenuated PD process. The chemical structures were drawn using ChemDraw software (version 18.0.0.231). Grey arrows mean promotion and grey blunt-ended bars mean inhibition.

**Table 1 ijms-25-01641-t001:** Inflammation-related genes associated with PD.

Treatments	Effects and Mechanisms	References
*LRRK2* mutant, human	IL-8 ↑, MCP-1 ↑, MIP-1-β ↑, BDNF ↑	[40]
C57BL/6J-Tg(LRRK2*R1441G)3IMjff/J mice, C57BL/6-Lrrk2 tm4.1 Arte mouse strain	DA neurons ↓, IL-6 ↑	[41]
G2019S mutation in the *LRRK2* gene, MPP^+^, induced pluripotent stem cells (iPSCs)	Neurites and neurite arborization ↓, survival rate ↓, apoptosis ↑, *IL-1β* ↑, *TNF-α* ↑, *COX-2* ↑, *IL-6* ↑, *iNOS* ↑	[42]
Macrophages and microglia from PD patients and mice with LRRK2-G2019S mutation	DA neurons ↓, WAVE2 stabilization ↑, phagocytic response ↑	[43]
AAV2 vectors harbor the cDNA encoding human α-synuclein, microglia, primary neurons	H_2_O_2_ ↑, microglial migration	[44]
α-Synuclein fibrils, primary microglia	IL-1β ↑, caspase-1 recruitment, NLRP3 activation ↑,	[45]
A53T mutant α-synuclein, microglia	oxidative stress, nicotinamide adenine dinucleotide phosphate oxidase activity ↑, P2X7 receptor ↑, PI3K/AKT ↑	[46]
A53T mutant α-synuclein, A30P mutant α-synuclein, microglia	NF-κB/AP-1/Nrf2 pathway activation ↑, TNF-α ↑, CXCL10 ↑	[47]
A53T mutant α-synuclein, Cx3cr1^−/−^ mice	DA neurons ↓, *RelA* ↑, *IL-1β* ↑, *TNF-α* ↑, *IL-6* ↑	[48]
LV-FLEX-*SNCA*^G420A^, C57BL/6 mice	DA neurons ↓, phagocytic exhaustion, ROS ↑, NO ↑, inflammasome pathway, inflammatory cytokine, and inflammatory nuclear transcriptional pathway gene enrichment	[49]
α-Synuclein, microglia	MMPs ↑, NO ↑, ROS ↑, IL-1β ↑, TNF-α ↑, PAR-1 activity ↑,	[50]
α-Synuclein, microglia, C57BL/6 mice	TNF-α ↑, TLR4 activation, TLR4/PI3K/AKT/GSK3β pathway ↑, NF-κB ↑	[51]
Exhaustive exercise, Parkin^−/−^ mice, Pink1^−/−^ mice	IL-6 ↑, IFNβ1 ↑, DA neurons ↓, circulating mtDNA ↑	[52]
LPS, siRNA-mediated downregulation of Parkin, Parkin^−/−^ mice, microglia	The vulnerability of DA neurons to inflammation ↑, survival of activated microglia ↑	[53,54]
*Pink1*/*Parkin* mutant, human	IL-6 ↑, ccf-mtDNA ↑, mitophagy ↓, ccf-mtDNA-cGAS-STING-IL-6 pathway ↑	[55]
Intestinal infection with Gram-negative bacteria, Pink1^−/−^ mice, primary neuron	Mitochondrial antigen presentation, establishment of cytotoxic mitochondria-specific CD8+T cells	[56]
LPS, *Parkin* mutant, iPSC-derived neuron–microglia co-cultures	Mitochondrial biogenesis pathway ↓, mtDNA dyshomeostasis, sirtuin 1 ↓, immune response ↑	[57]
LPS, Pink1^−/−^ mice	*IL-1β* ↑, *TNF-α* ↑, *IL-6* ↑, IL-1β ↑, TNF-α ↑, IL-6 ↑, Ca^2+^ storage capacity ↓, JNK activity ↑, DA level ↓	[58,59]
Pink1^−/−^ mouse embryonic fibroblasts	Autodimerization and autoubiquitination of TRAF6 ↑, polyubiquitination of TAK1 ↑, NF-κB activity ↑	[60]
*Parkin* KO mice, astrocyte	NOD2 ↑, ER stress ↑, JNK activation ↑, cytokine release ↑, neurotropic factor ↓	[61]
*GBA* mutant, human	Microglia activation	[62]
LPS, OF1 strain mice, zebrafish, GD iPSC-derived neuronal progenitor cells	Microglia activation, gene and protein expression of members of the Wnt/β-catenin signaling pathway ↓	[63,64]
*GBA* mutant, human	Cognitive dysfunction ↑, IL-8 ↑, monocyte chemotactic protein 1 ↑, macrophage inflammatory protein 1α ↑	[65]
*GBA* mutant, GCase inhibitor CBE, long-lived Gaucher mice	Glial activation, α-synuclein accumulation	[66]
GCase inhibitor CBE, BDF1 mice	α-synuclein aggregates, neuroinflammation ↑, complement C1q ↑, abnormalities in synaptic, axonal transport, and cytoskeletal proteins	[67]
GCase inhibitor CBE, C57BL/6N mice	Enhanced MPTP-induced neurodegeneration, α-synuclein ↑, microglia activation	[68]
Dj-1^−/−^ astrocytes, primary neuron–astrocyte co-culture	NO ↑, iNOS ↑, p38 MAPK phosphorylation ↑, COX-2 ↑, IL-6 ↑, apoptosis ↑	[69]
Increased astroglial DJ-1 expression, zebrafish, astrocyte	Protected from oxidative-stress-induced injuries as caused by MPP^+^	[70]
Rotenone, overexpress human DJ-1 protein, Lewis rats	Astrocytes with DJ-1 overexpression showed a marked reduction in neuronal oxidative stress and microglial activation, α-synuclein accumulation and phosphorylation were decreased within DJ-1-transduced animals	[71]
DJ-1 depletion, microglia	IL-1β ↑, IL-6 ↑, monoamine oxidase activity ↑, ROS ↑, NO ↑, TREM2 ↓	[72]
DJ-1 knockdown, C57BL/6 mice, BV2 murine microglial cells	Neuroinflammation ↑, apoptosis ↑, DA neurons ↓, Nrf2 ↓, NLRP3 ↑	[73]
DJ-1 knockout, C57BL/6 mice, primary astrocytes	TNF-α ↑, HO-1 ↓, PTGDS levels increased in the injured brain of WT mice, but barely in that of knockdown mice	[74]

↑ represents upregulation, ↓ represents downregulation.

**Table 2 ijms-25-01641-t002:** A summary of the potential anti-inflammatory therapies for PD.

Strategies	Chemical Structure	Treatments	Effects and Mechanisms	References
NLRP3 inflammasome inhibition		Antrodia camphorata polysaccharide, 6-OHDA, MES23.5	ROS-NLRP3 ↓, apoptosis of dopaminergic neurons ↓, dopamine ↑	[145,146]
	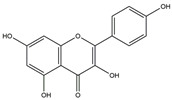	Kaempferol, LPS, C57 BL/6J mice, SD rats	NLRP3 inflammasome activation ↓, caspase-1 ↓, NLRP3-PYCARD-CASP1 complex assembly ↓, macroautophagy/autophagy ↑, the activation of microglia and astrocytes ↓, TH-positive neurons loss↓, IL-1β ↓, IL-18 ↓, iNOS ↓, COX-2 ↓, p38MAPK/NF-κB ↓	[147,148]
	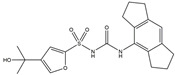	MCC950, LPS, primary microglia	NLRP3 inflammasome activation ↓, α-synuclein-mediated inflammasome activation ↓	[149]
	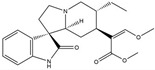	Rhynchophylline, MPTP, C57 BL/6J mice	*TLR4* ↓, *NLRP3* ↓, *COX2* ↓,	[150]
	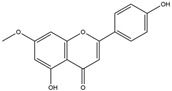	Genkwanin, MPP+, SH-SY5Y cells	Cell viability ↑, LDH release ↓, apoptosis ↓, ROS generation ↓, SOD activity ↑	[151]
	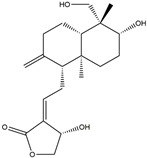	Andrographolide, microglia, MPP+, LPS	NLRP3 inflammasome activation ↓, ATP level ↑, dopaminergic neuron ↑, behavioral parameter improvement	[152]
	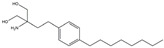	Fingolimod, MPTP, C57 BL/6J mice	Behavioral deficits ↓, DA neurons loss ↓, dopamine levels ↑, microglial activation ↓, IL-6 ↓, IL-1β ↓, TNF-α ↓	[153]
	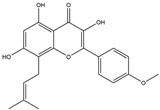	Icaritin and its glucoside, MPTP, 6-OH-DOPA, C57 BL/6 mice, Nrf2 knockout mice	NLRP3 inflammasome activation ↓, IL-1β ↓, VDAC and ATP5B stabilization, Nrf2 signaling activation ↑	[154,155]
	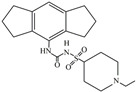	Selnoflast, human	NLRP3 inflammasome activation ↓, the trial is being tested	[246]
NF-κB inhibition	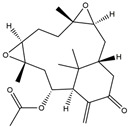	Hypoestoxide, mThy1-α-syn transgenic mice	Decreased microgliosis, astrogliosis, and pro-inflammatory cytokine gene expression, DA neurons ↑, α-synuclein pathology ↓, phosphorylated NF-κB ↓	[156]
	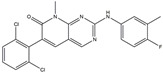	PD180970, MPTP, C57 BL/6J mice	IL-6 ↓, MCP-1 ↓, NF-κB ↓	[157]
	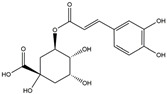	Chlorogenic acid, MPTP, Swiss albino male mice	Neuroinflammation ↓, TNF-α ↓, IL-1β ↓, IL-10 ↑, attenuation of astrocyte activation	[158]
		Tea extracts, 6-OHDA, PC12 cells, SH-SY5Y cells	NF-κB nuclear translocation and binding activity ↓	[159]
	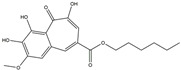	CU-CPT22, oligomeric α-synuclein, primary microglia	Nuclear translocation of NF-κB ↓, TNF-α ↓	[118]
	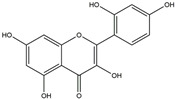	Morin, MPP+, PC12 cells, C57 BL/6 mice	Cell viability loss and apoptosis ↓, behavioral deficits ↓, DA neurons ↑, dopamine depletion ↓, astrocyte activation ↓, ROS ↓	[160,161]
	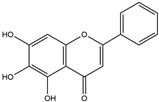	Baicalein, MPTP, C57 BL/6 mice	Motor ability ↑, DA neuron loss ↓, microglial and astrocyte activation ↓, nuclear translocation of NF-κB ↓, activation of JNK and ERK ↓	[162]
	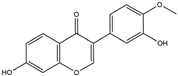	Calycosin, MPTP, LPS, mice, BV2 cells	Behavioral dysfunctions ↓, inflammatory responses ↓, TLR/NF-κB and MAPK pathways ↓	[163]
	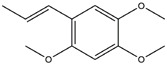	α-Asarone, MPTP, LPS, C57 BL/6 mice, BV-2 cells	Neuroinflammatory responses ↓, pro-inflammatory cytokine ↓, microglial activation ↓, behavioral impairments ↓	[164]
	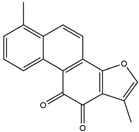	Tanshinone I, LPS, MPTP, BV-2 cells, C57 BL/6 mice	NO ↓, TNF-α ↓, IL-1β ↓, IL-6 ↓, granulocyte colony-stimulating factor expression ↓, NF-κB activation ↓, improved motor functions, normalized striatal neurotransmitters	[165]
	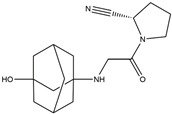	Vildagliptin, rotenone, Wistar rats	Neuronal demise ↓, cytochrome c ↓, caspase-3 ↓, RAGE/NFκB cascade ↓	[166]
	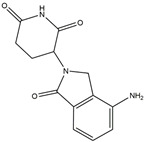	Lenalidomide, mThy1-α-syn transgenic mice	Behavioral deficits ↓, DA fiber loss ↓, microgliosis ↓, pro-inflammatory cytokines ↓, NF-κB activation ↓	[167]
Microglia inhibition	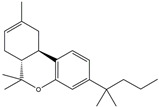	JWH133, MPTP, C57 BL/6 mice	BBB damage ↓, infiltration of peripheral immune cells ↓, iNOS ↓, pro-inflammatory cytokines ↓, microglial activation ↓	[168]
	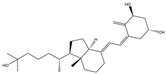	1,25-dihydroxyvitamin D3, 6-OHDA, MPTP, SD rats, C57 BL/6 mice	Microglial activation ↓, pro-inflammatory cytokine expression ↓	[169]
	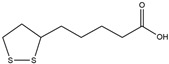	α-Lipoic acid, MPTP, C57 BL/6 mice	Step length and suspension time ↑, microglial activation ↓, NF-κB ↓, TNF-α ↓, iNOS ↓	[170]
	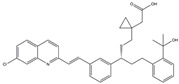	Montelukast, 6-OHDA, rotenone, C57 BL/6 mice, Wistar rats	Neuroinflammatory activities ↓, TNF-α ↓, IL-1β ↓, microglial activation ↓, p38 MAPK ↓, CysLT1 ↓	[171,172]
	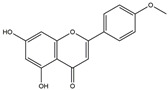	Acacetin, MPTP, primary mesencephalic cells	DA neuron loss ↓, NO ↓, PGE2 ↓, TNF-α ↓, time of turning and locomotor activity ↓, dopamine level ↑, microglial activation ↓	[173]
	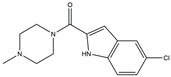	JNJ7777120, rotenone, SD rats	Microglial markers ↓, dopaminergic neuron degeneration ↓	[174]
	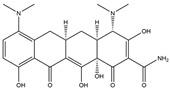	Minocycline, LPS, Wistar rats	IL-1α ↓, TNF-α ↓, OX-6 ↓, OX-42 ↓, 3-nitrotyrosine immunoreactivity ↓	[176]
	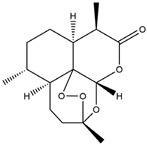	Artemisinin, MPTP, C57 mice	Improved behavioral symptoms, DA neurons ↑, microglial activation ↓	[177]
	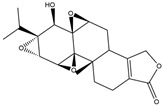	Triptolide, LPS, BV2 cells, primary microglia	mGlu5 expression ↑, mGlu5 membrane localization ↑, NO ↓, iNOS ↓, TNF-α ↓, IL-1β ↓, IL-6 ↓, microglial activation ↓	[179]
		NXP031, MPTP, C57 BL/6J mice	DA neuron loss ↓, motor impairment ↓, plasma ascorbic acid levels ↑, microglia activation-induced neuroinflammation ↓	[180]
	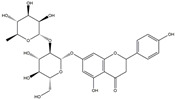	Naringin, 6-OHDA, C57 BL/6 mice	mTORC1 activation ↑, microglial activation ↓, DA degeneration ↓	[181]
	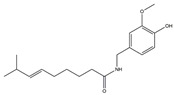	Capsaicin, LPS, SD rats	DA neuron loss ↓, pro-inflammatory mediators ↓, shift the pro-inflammatory M1 microglia/macrophage population to an anti-inflammatory M2 state, iNOS ↓, arginase 1 ↑, peroxynitrate production ↓, ROS ↓, oxidative damage ↓	[182]
	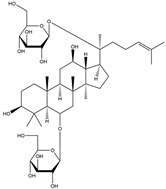	Ginsenoside Rg1, MPTP, C57 BL/6J mice	TH-positive cells ↑, TNF-α ↓, IFN-γ ↓, IL-1β ↓, IL-6 ↓, microglial activation ↓, infiltration of CD3+ T cells into the SNpc region ↓	[183]
		Vasoactive intestinal peptide, MPTP, 6-OHDA, α-synuclein, C57 BL/6J mice, SD rats	Microglial inflammatory responses ↓, robust nigrostriatal protection, increased neuronal sparing, astrogliosis ↓, inflammatory microglia ↓, DA neurons ↑, improved striatal densities	[186,187,188,189]
Astrocyte inhibition	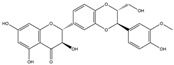	Silibinin, MPTP, C57 BL/6 mice	Motor dysfunction ↓, DA neuron loss ↓, glial activation ↓, ERK and JNK phosphorylation ↓	[190]
		Bear bile powder, MPTP, C57 BL/6 mice	Dyskinesia ↓, tyrosine hydroxylase ↑, astrocyte hyperactivation ↓, glial fibrillary acidic protein ↓, COX2 ↓, iNOS ↓, TGR5 ↑, phosphorylation of protein kinase B ↓, NF-κB ↓	[191]
NADPH oxidase inhibition	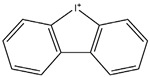	Diphenyleneiodonium, LPS, MPP+, rotenone, neuron–glia cultures	DA neurons ↑, NADPH oxidase activation ↓, superoxide production ↓	[193]
	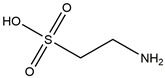	Taurine, paraquat and maneb, C57 BL/6J mice	Microglial M1 polarization ↓, gene expression levels of pro-inflammatory factors ↓, NADPH oxidase ↓, NF-κB ↓	[194]
PPARγ agonist	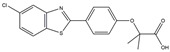	MHY908, MPTP, C57 BL/6 mice, SH-SY5Y neuroblastoma cells, primary astrocytes	Inflammatory response ↓, DA neuron loss ↓, motor deficit ↓, glial activation ↓, ROS ↓	[195]
	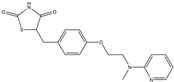	Rosiglitazone, MPTP, C57 BL/6J mice	M2 anti-inflammatory microglia were stimulated and inflammatory microglia were inhibited	[196]
	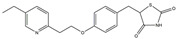	Pioglitazone, MPTP, rhesus monkeys, C57 BL/6 mice	DA neuron loss ↓, DA levels ↑, microglial activation ↓, NF-κB ↓	[197,198]
Targeting MAPK pathway	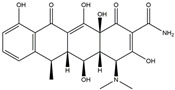	Doxycycline, 6-OHDA, LPS, C57BL/6 mice, primary microglial cell	DA neuron loss ↓, microglial activation marker IBA-1 ↓, ROS ↓, NO ↓, pro-inflammatory cytokines ↓, phosphorylation level of MAPK ↓	[199,200]
	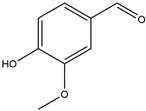	Vanillin, LPS, Wistar rats	Motor dysfunction ↓, DA neuron loss ↓, microglial activation ↓, iNOS ↓, COX-2 ↓, IL-1β ↓, IL-6 ↓, p-ERK1/2 ↓, p-p38 ↓	[201]
	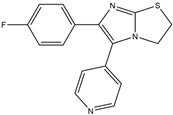	SKF-86002, mice overexpressing α-synuclein	Neuroinflammation ↓, ameliorated synaptic, neurodegenerative, and motor behavioral deficits, α-synuclein accumulation ↓, p38α/β ↓	[202]
Targeting AMPK-dependent pathway	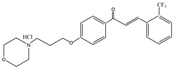	KMS-99220, LPS, BV2 cells	Phosphorylation of IκB ↓, nuclear translocation of NF-κB ↓, iNOS ↓, NO ↓	[203]
		Liraglutide, MPTP, C57 BL/6 mice	Iba1 and GFAP expression ↓, p-AMPK ↑, NF-κB ↓, neuroinflammation ↓	[204]
	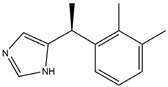	Dexmedetomidine, MPTP, C57BL/6J mice	Motor dysfunction ↓, mechanical allodynia ↓, thermal hyperalgesia ↓, astrocyte activation ↓, TNFα and IL-6 expression ↓, p-AMPK ↑, mTOR ↓, NF-κB ↑	[205]
	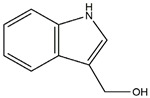	Indole-3-carbinol, rotenone, SD rats	Motor dysfunction ↓, striatal DA decrease ↓, weight loss ↓, TH expression reduction ↓, α-synuclein ↓, SIRT1-AMPK signaling pathway ↑	[206]
		Healthy human fecal microbiota Transplantation, MPTP, C57BL/6J mice	Motor impairments ↓, DA neurodegeneration ↓, glial activation ↓, colonic inflammation ↓, AMPK/SOD2 signaling pathway ↑	[207]
Targeting α-synuclein		Anti-TLR2, α-synuclein-overexpressing transgenic mice	α-Synuclein accumulation ↓, neuroinflammation ↓, neurodegeneration ↓, behavioral deficits ↓	[208]
		PD03A, PD01A, patients with PD	A good safety and tolerability profile, humoral immune response against the α-synuclein target epitope	[209,210]
	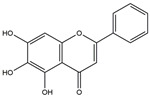	Baicalein, MPP+, SD rats	Dopamine ↑, α-synuclein aggregates ↓, ED-1 ↓, caspase-1 ↓, IL-1β ↓, cathepsin B ↓	[211]
		PRX002, human	Favorable safety, tolerability, and pharmacokinetic profiles at all doses tested, with no immunogenicity, serum α-synuclein ↓	[247]
Targeting miRNAs		miR-7, MPTP, α-synuclein PFF or PBS injection, C57BL/6J mice	α-Synuclein ↓, oxidative stress ↓, less pronounced degeneration of the nigrostriatal pathway, better behavioral performance, neuroinflammatory reaction ↓, NLRP3 inflammasome activation ↓, DA neuron degeneration ↓, microglial activation ↓	[212,213,214]
	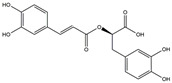	Rosmarinic acid, MPTP, C57BL/6 mice	Motor function improvement, inflammatory responses ↓	[215]
		MicroRNA-3473b inhibitor, LPS, MPTP, BV2 cells, C57BL/6J mice	Secretion of inflammatory factors ↓, autophagy ↑, microglial activation ↓, TREM2/ULK1 ↑	[216]
Acupuncture		Acupuncture, MPTP, C57BL/6 mice	MAC-1 ↓, COX-2 ↓, iNOS ↓, motor function improvement, comorbid anxiety ↓, DA fibers and neurons ↓, activation of microglia and astrocyte ↓, conversion of Bax and Bcl-2 expression, physiology function restoration, neuroinflammation ↓	[217,218]
Exercise		Exercise, human, MPTP, C57BL/6 mice	Dendritic spine density ↑, arborization ↑, synaptic protein PSD-95 ↑, synaptophysin ↑, cognition improvement, motor function restoration, pro-inflammatory cytokine ↓, α-synuclein ↓, TLR2 activation ↓, autophagy ↑, mitochondrial function improvement, Sirt1 pathway ↑, DA neuron loss ↓, apoptosis ↓, mitochondrial biogenesis ↑, oxidative stress ↓, PGC-1α ↑	[219,220,221,222,223]
		Treadmill exercise, endurance exercise, ICR mice, albino mice	DA neuron loss ↓, motor balance ↑, coordination dysfunction ↓, iNOS ↓, MAPKs ↓, microglial activation ↓, nNOS ↓	[224,225]
		Exercise, MPTP, C57BL/6 mice	Motor deficits ↓, neurogenesis ↑, DA neuron loss ↓, antioxidant capacity ↑, autophagy ↑	[227]
Others		Anti-TNF, human	Incidence of PD in patients with IBD ↓	[228]
		Exendin, C57 BL/6 mice, MPTP	Microglial activation ↓, MMP3 ↓, pro-inflammatory molecules ↓	[229,230]
	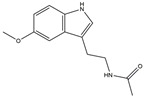	Melatonin, C57 BL/6 mice, MPTP	COX-2 activity ↓, lipid peroxides ↓, nitrite/nitrate ↓	[232]
	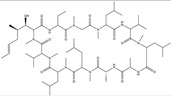	Cyclosporin, SD rats, AAV-α-synuclein	Motor performance ↑, NFATc3 ↓, mitochondrial stress ↓	[233]
		Probiotics, patients with PD	Abdominal pain ↓, bloating ↓, bradykinesia ↓, rigidity ↓, tremor ↓, gait dysfunction ↓, inflammation ↓	[235,236]
		Nonsteroidal anti-inflammatory drugs, patients with PD	PD risk ↓	[237]
	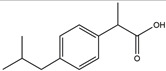	Ibuprofen, C57 BL mice, human, MPTP	Dopamine level ↑, ROS ↓, COX inhibition, PD risk ↓	[238,248]
	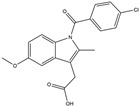	Indomethacin, C57 BL mice, MPTP	Microglial activation ↓, lymphocytic infiltration ↓	[239]
	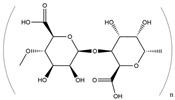	Polymannuronic acid, C57 BL/6J mice, MPTP	Improved motor functions, DA neuronal loss ↓, HVA ↑, 5-HT ↑, 5-HIAA ↑, GABA ↑, pro-inflammatory cytokines ↓, MAPK ↓	[240]
	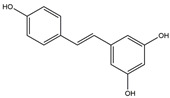	Resveratrol, Wistar albino rats, SD rats, 6-OHDA, rotenone	ER stress ↓, CHOP ↓, GRP78 ↓, caspase-3 activity ↓, glutathione peroxidase ↑, Nrf2 ↑, ROS ↓, COX-2 ↓, PLA2 ↓, TNF-α ↓, neuroinflammation ↓	[241,242]
		Curcuminoids, C57 BL/6 mice, MPTP	Dopamine depletion ↓, iNOS ↓, GFAP ↓, pro-inflammatory cytokine ↓, total nitrite generation ↓, motor performance improvement, gross behavioral activity improvement	[243]
	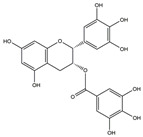	EGCG, Wistar rats, C57 BL/6 mice, Lewis rats, rotenone, MPTP	Motor impairments ↓, NO level ↓, LPO ↓, SDH activity ↑, ATPase activity ↑, ETC enzyme activity ↑, catecholamines ↑, neuroinflammatory and apoptotic markers ↓, mitochondrial dysfunction ↓, neurochemical deficiency ↓, TH activity ↑	[242,244]
	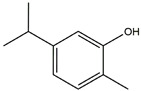	Carvacrol, Wistar rats, 6-OHDA	TNF-α ↓, IL-1β ↓, DA neuron loss ↓	[245]
		DNL151, human	Delay the worsening of pathology for patients with early-stage PD	[249]
		Sargramostim, human	PD improvement	[250]
		Exenatide, human	Exenatide suppress motor impairments and improves cognitive function of patients with PD	[251]
	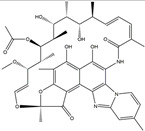	Rifaximin, human, transgenic PD mice (MitoPark)	Relative abundance of Flavonifractor ↑, anti-inflammatory effect ↑	[252]

↑ represents upregulation, ↓ represents downregulation. The chemical structures were drawn using ChemDraw software (version 18.0.0.231).

## Data Availability

Not applicable.

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
