# Peer review of "What Can Inflammation Tell Us about Therapeutic Strategies for Parkinson’s Disease?"

_ijms, 2024, doi:10.3390/ijms25031641_

Round 1
Reviewer 1 Report
Comments and Suggestions for Authors
This paper deals with an interesting and complex aspect of the pathophysiology of Parkinson's disease. The paper is well written and covers all the most important aspects. I would suggest adding a brief summary in the introduction section about the role of inflammation in the pathophysiology of PD and PD symptoms (for example: doi:10.3233/JPD-202417, doi:10.1016/j.bbi.2020.07.005) in order to give an important clinical impact of the reported aspects.
Reviewer 2 Report
Comments and Suggestions for Authors
This review overviews a vast pieces of research about the inflammation genes and immune signaling pathways associated with PD suggesting inflammation-mediated therapies. The enough numbers of papers are quoted, well summarized using Tables and well interpreted. Thus I think this review is interesting and helpful to study about complex PD etiology, pathology and therapeutics in the light of immunological aspects.
Reviewer 3 Report
Comments and Suggestions for Authors
The authors wrote a review focussing on the impact of inflammation Parkinson’s disease and on therapeutic strategies to reduce inflammation. Authors want to clarify the role of neuroinflammation in PD and provide directions for future research.
The review is well written and gives a comprehensive overview of the very many factors and their genes found and interplaying in PDs inflammation. Also, possible inflammation-related therapies according to their mode of action in PD are dealt with. The figures are clear and instructive.
Of course, it is not possible for the reviewer (me) to read the manuscript word by word or even have a look on at least some of the 245 references and their meaningfulness. It is completely out of the scope of a review to “control” the tables or look for possible mistakes.
In the discussion the authors claimed: Therefore, it’s urgent to find more effective and safer therapeutic strategies. Inflammation is an early event in PD progression and plays a vital role in PD pathogenesis. What I am missing is a short statement which of the many factors described and their dependencies are already addressed by drugs (for instance effectivness in clinical trials) influencing the factors in PD patients, especially in early stages of PD.
